# Ecology and Diversity of Angiosperm Parasites and Their Host Plants along Elevation Gradient in Al-Baha Region, Saudi Arabia

Sami Asir Al-Robai

Department of Biology, Faculty of Science, Al-Baha University, Al-Baha 1988, Saudi Arabia; salrobai@bu.edu.com

**Abstract:** The ecology and diversity of flowering parasitic plants and their hosts are poorly investigated and usually ignored in Saudi Arabian plant communities. Therefore, this work aimed at assessing the ecology and diversity of parasitic plants and their hosts along an elevation gradient in the Al-Baha region (1300–2400 m.a.s.l.). Different quantitative vegetation parameters were applied to analyze the collected data. Eight parasitic plants from six genera and four families were identified along the gradient, with 67% of them being zoochorously dispersed species. They accounted for approximately 23.5% (8 out of 34) of those found throughout Saudi Arabia. Perennials, stem hemi-parasites, and biregional taxa accounted for around 62.5% of the total parasites, whereas indigenous species accounted for 75%. The dominant family of parasitic species was Loranthaceae (50%), and *Phragmanthera austroarabica* A.G.Mill. & J.A.Nyberg was the most important species (IVI = 107.28). *Orobanche cernua* Loefl. and *Loranthella deflersii* (Tiegh.) S.Blanco & C.E.Wetzel were restricted to the dry zone (low elevation) only, while the other parasites were distributed across the surveyed region. Twenty-three host plants were identified throughout the study region. About 83% of them were phanerophytes and bioregional plants, with 91% being perennial species. The prevalent host plant family across all sites was Fabaceae, with *Nicotina glauca* Graham being the most important host species (IVI = 32.44%). *P. austroarabica* and *Plicosepalus curviflorus* Tiegh. preferred *Vachellias* as host plants, while *Vachellia flava* (Forssk.) Kyal. & Boatwr. was the heavily infected host by *P. austroarabica*. *P. austroarabica* had a broad spectrum of host range (13 host plants), while *O. cernua* had a very narrow host range (only *Rumex nervosus* Vahl). Individual parasite and host species were markedly more abundant in the wet zone than in the low-altitude dry zone. Further research is needed to fully understand such distinctive groups of plants and their negative and positive ecological consequences on plant biodiversity and natural ecosystems.

**Keywords:** parasitic plants; host species; elevation; vegetation parameters; diversity; Saudi Arabia

## 1. Introduction

Parasitic angiosperms are a diverse group of around 4500 species divided into 12 families and roughly 300 genera [1]. They are specialized plants that lose the capacity to photosynthesize entirely or partially and receive their organic and inorganic nutrients from the host plants through the haustoria. A parasitic plant's haustorium is a specialized structure that has a physiological connection with the host plants [2]. Although the number and diversity of these parasites vary depending on the biome and ecosystems, they are common elements in terrestrial environmental habitats worldwide [3]. Beyond having a bad impact on the host, their ecological roles are much more nuanced [4]. Some parasitic species can change the community competition patterns, facilitate the environmental cycling of nutrients [5], and modify diversity within communities [6]. It has been frequently shown that parasitic plants can inhibit the development and competitiveness of widespread plants, lower the rate of biomass formation in the ecosystem, and promote seed growth by creating openings for embryos to germinate [7,8]. This means that the

overall harm caused by parasitic plants to their hosts could be changed into a benefit for wider plant communities [4]. Some of them play significant roles in controlling plant invasions and aiding in biodiversity restoration [9,10]. Furthermore, the haustorial link may aid in the flow of stress-responsive chemicals as well as potentially hazardous substances such as heavy metals [11,12]. However, a small number of parasitic plants are among the most devastating agricultural pests, costing billions of dollars in annual losses, food poverty, and ecological risks [13]. Mistletoe is an airborne parasitic plant that is pollinated and disseminated by birds and other visitors; it controls the natural structure of the plant groups throughout the local habitat [14].

Depending on their photosynthetic ability, they can be classified as holoparasites (photosynthetic) or hemiparasites (non-photosynthetic) [15] or as stem or root parasites according to where they attach [16]. Although the haustorium structure varies throughout parasitic plant families, it is a feature common to all species [16].

Orobanchaceae and the order Santalales have the highest number of parasitic plants [2], with Loranthaceae (76 genera, more than 1000 species) having most of them [17]. Loranthaceae is predominantly distributed in Asia, the Americas, Africa, and Australia, with certain species spreading in areas having mild temperatures in Europe and East Asia [18]. It is primarily composed of airborne parasitic plants but also includes three root parasitic species [18]. The plants in this family interact with insects, birds, and mammals and have a crucial role in the biological ecosystems in which they exist [14]. The family *Loranthaceae* has four genera, *Phragmanthera*, *Oncocalyx*, *Tapinanthus*, and *Plicosepalus*, all of which grow natively in Saudi Arabia, with six species belonging to these genera dispersed across the Kingdom's north, west, and south [19]. There have been few publications, most of which have focused on species in a relatively restricted area of the Al-Baha region [20–22]. These publications barely mention the distribution and host range of parasitic vascular plants in this region. A few studies [23,24] have been conducted, but they do not address the ecology and diversity of all parasitic plants and their host plants that are present throughout the country. Consequently, this work gives an in-depth investigation on the distribution and diversity of parasitic plants and their host range of flowering plants in the Al-Baha region.

## 2. Materials and Methods

### 2.1. Study Area

Al-Baha is a highland region in southwestern Saudi Arabia lying at the longitude 41/42 E and latitude 16/20 N with an elevation ranging from 1300 to 2450 m above sea level (Figure 1). It is bordered on the west by the Rocky Mountains and on the east by semi-arid mountains and has a broad diversity of habitats associated with different plant species. The majority of Al-Baha's region falls within the tropical and subtropical arid zones [20]. Summer temperatures range from 22 to 32 °C and winter temperatures range from 10 to 22 °C, and the rainfall in arid areas varies between 100 and 200 mm [25]. Alaqiq, Baljurashi, Al Mandaq, and Al Mikhwah areas receive annual precipitations of 142, 300, 316, and 200 mm, respectively [20].

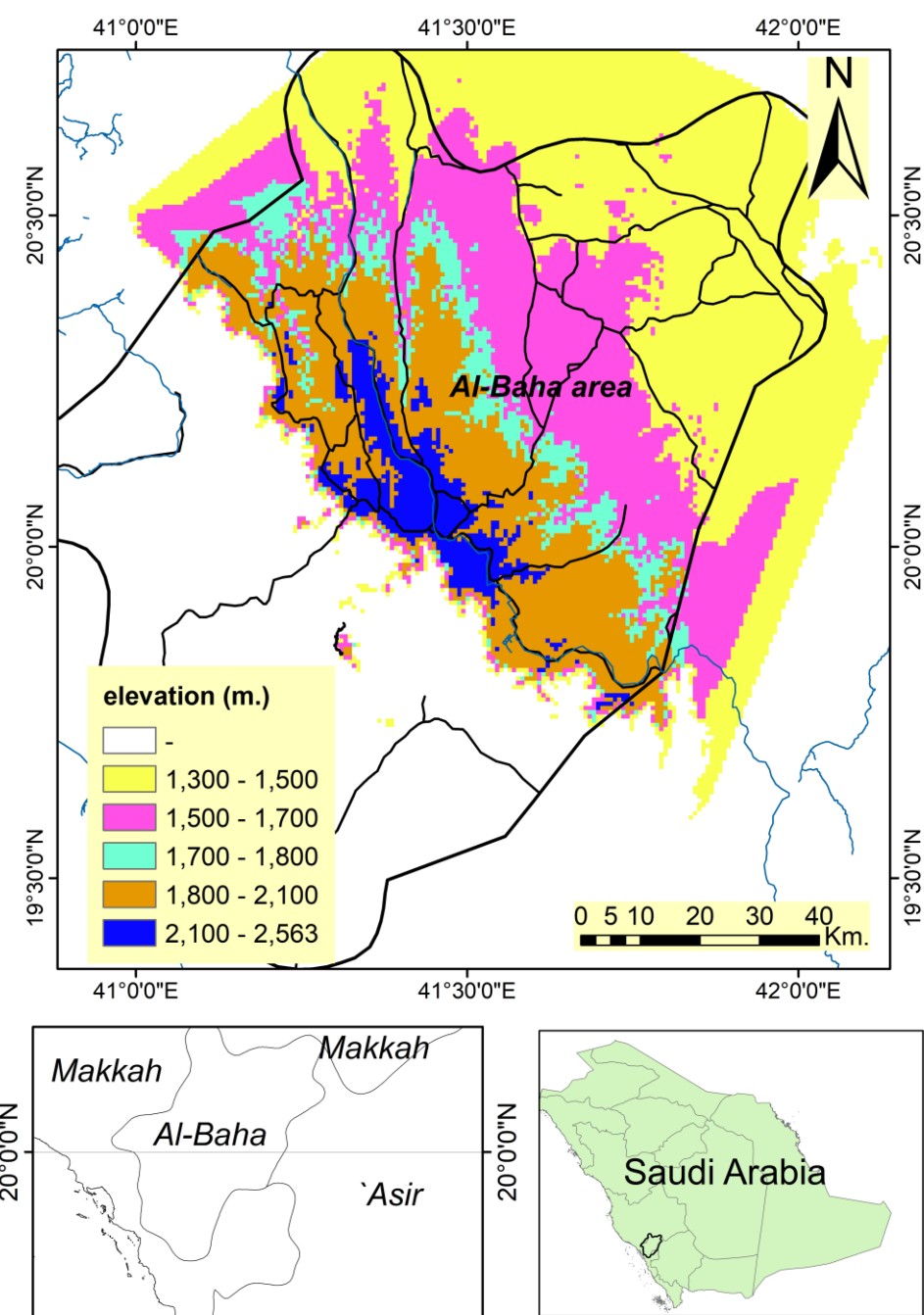

**Figure 1.** A map shows the locations of the study area.

*2.2. Field Survey*

Extensive field surveys (over 30 trips) were conducted to cover all the sites of the study region from April 2022 to March 2023. The region was divided into two zones, dry (low altitude) and wet (high altitude) zones. According to the elevation gradient, each zone was subdivided into two sites, dry zone (site 1: 1300–1500, site 2: 1501–1700 m.a.s.l.) and wet zone (site 3: 1800–2100, site 4: 2101–2400 m.a.s.l.). Further, each site was divided into eight stands with 10 quadrats (20 × 20 m). An overall 80 quadrat units were analyzed in each site. Plant samples were collected once, and there was no seasonal sampling.

*2.3. Plant Collection and Species Identification*

The examined plant taxa were identified and revised using different published volumes of Saudi Arabia flora [26–29]. Nomenclature was reviewed and updated using

authenticated international data from the World Online [30]. Voucher specimen from each plant was deposited at Biology Department, Faculty of Science, Al-Baha University.

### 2.4. Floristic Analysis

Life form, habit, and life span categories were determined using the updated Raunki-aer [31] classification by Govaerts et al. [32]. Phytogeographical categories were recognized following Wickens [33] and Zohary [34].

### 2.5. Data Analysis

Quantitative measurements derived from the quadrat method were performed to determine the relative frequency, relative abundance, and relative density of each species in the surveyed region. From these parameters, the species importance value index (IVI) was calculated to assess the dominant species in the study region [35,36]. The importance value defines the importance of a species in the community structure or species composition in the study area [37].

$$\text{Density (D)} = \frac{\text{Total number of individuals of a species in all quadrats}}{\text{Total number of quadrats studied}} \times 100$$

$$\text{Frequency (F)} = \frac{\text{Total number of quadrats in which species occur}}{\text{Total number of quadrats studied}} \times 100$$

$$\text{Abundance (A)} = \frac{\text{Total number of individuals of a species in all quadrats}}{\text{Total number of quadrats in which species occur}} \times 100$$

$$\text{\% Relative density (RD)} = \frac{\text{Density of the species}}{\text{Total density of all species}} \times 100$$

$$\text{\% Relative frequency (RF)} = \frac{\text{Frequency of the species}}{\text{Total frequency of all species}} \times 100$$

$$\text{\% Relative abundance (RA)} = \frac{\text{Total number of individuals of a species in all quadrats}}{\text{Total abundance of all species}} \times 100$$

$$\text{Importance value index (IVI)} = \text{RD} + \text{RF} + \text{RA}$$

Microsoft Excel was utilized as a software package (16.0) to analyze data and construct the charts and histograms that were used to present the collected data.

## 3. Results

### 3.1. Floristic Composition of Parasitic Species

Eight species of parasitic angiosperm plants were recorded in the studied four sites, namely, *Phragmanthera austroarabica* A.G.Mill. & J.A.Nyberg, *Viscum schimperi* Engl., *Cuscuta campestris* Yunck., *Orobanche mutelii* F.W.Schultz, *Orobanche cernua* Loefl., *Loranthella deflersii* (Tiegh.) S.Blanco & C.E.Wetzel, *Plicosepalus acaciae* (Zucc.) Wiens & Polhill, and *Plicosepalus curviflorus* Tiegh. (Table A1 and Figure 2). These species belong to Loranthaceae (four species), Orobanchaceae (two species), Santalaceae (one species), and Convolvulaceae (one species). All parasitic species were found in the two surveyed dry and wet zones, except for *O. cernua* and *L. deflersii* which were reported in the wet zone only (Figure 3). The total number of individual parasitic species ranged between 392 and 405 in the wet zone and 172 and 190 in the dry zone. *P. austroarabica*, *P. curviflorus*, and *V. schimperi* were the most abundant parasitic species in the study area (Figure 4). According to the WFO [38], *C. campestris* and *O. mutelii* were exotic plants while the remaining species were indigenous plants.

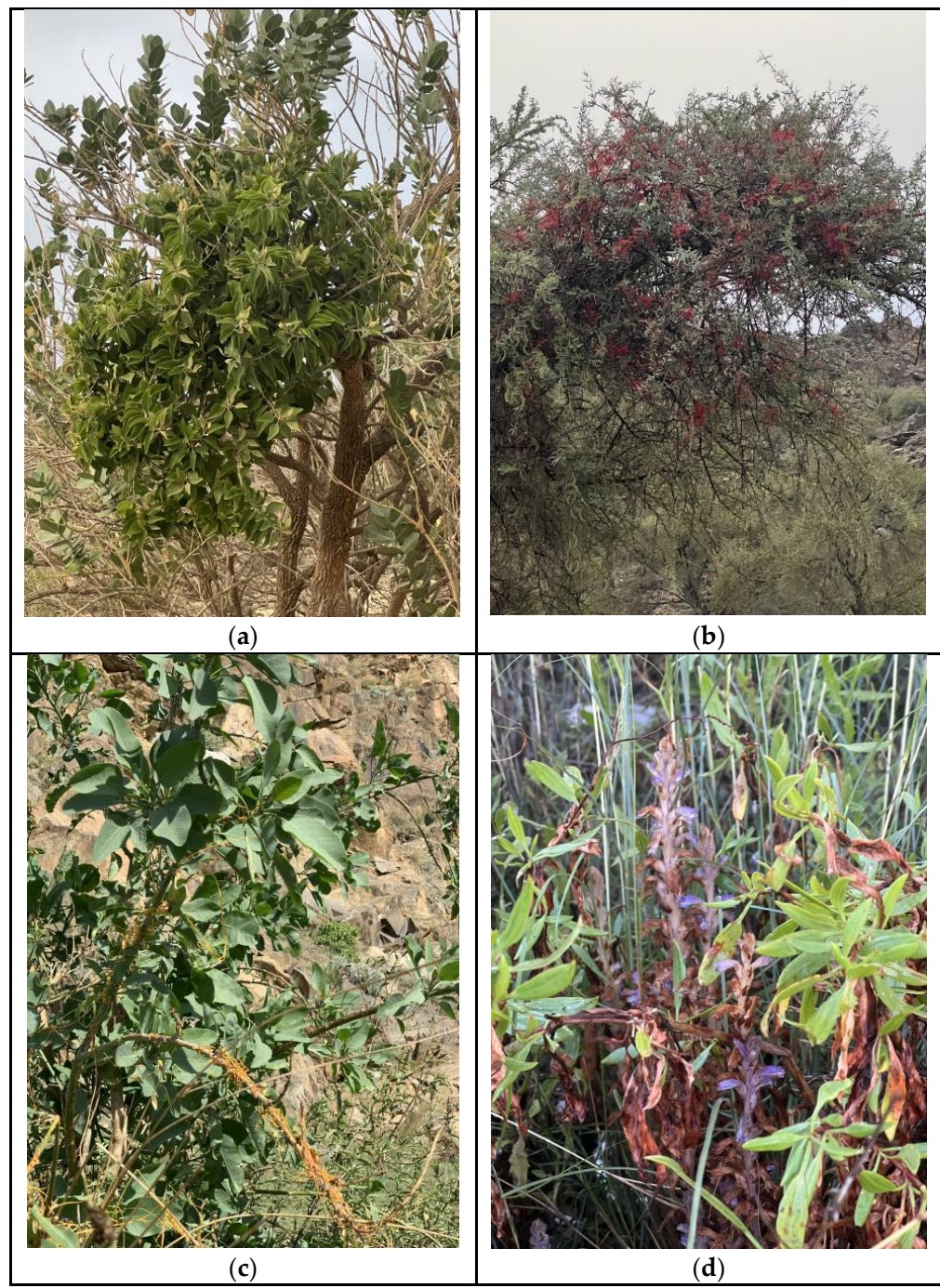

**Figure 2.** Some parasitic species and their host plants. (**a**) *P. austroarabica* on *Calotropis procera*. (**b**) *P. curviflorus* on *Vachellia gerrardii*. (**c**) *C. campestris* on *N. glauca*. (**d**) *O. cernua* on *Rumex nervosus*.

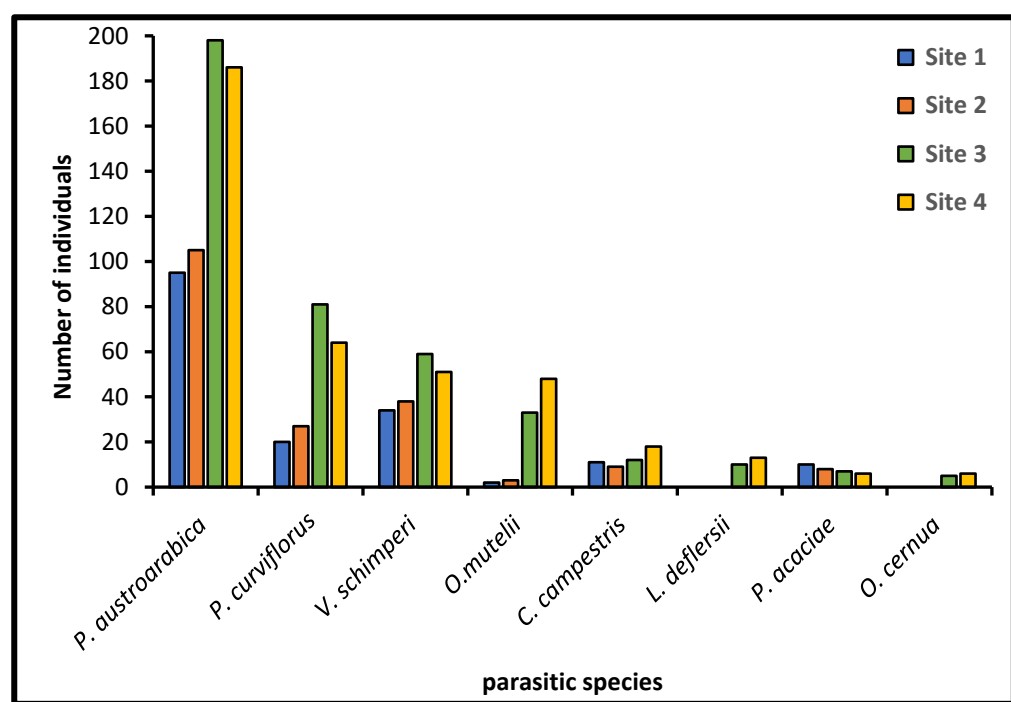

**Figure 3.** The number of individual parasitic species identified in each site of the study region.

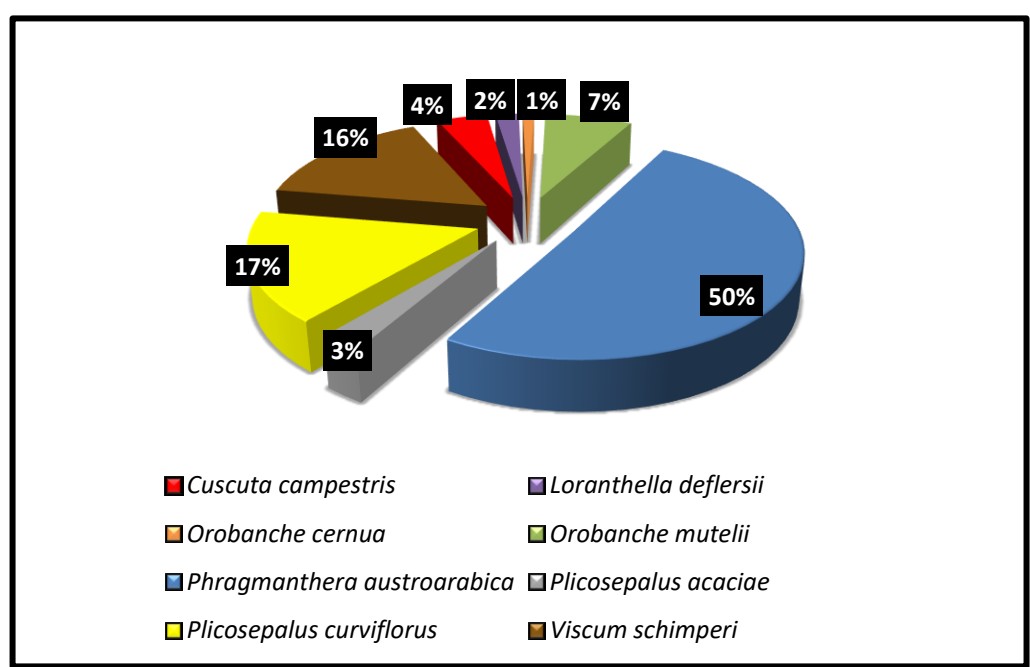

**Figure 4.** Percentage of the total number of individual parasitic species in the study region.

The identified angiosperm parasites can be categorized into four kinds: root hemiparasites, root holoparasites, stem hemiparasites, and stem holoparasites (Table A1). Most of the parasitic species belonged to stem hemiparasites (five species), followed by root holoparasites (two species), and stem holoparasites (one species). Of the total, five species (62.5%) were perennials/shrubs and three species were annual/herbs (37.5%).

The number of the host taxa that were attacked by parasitic species in the study area ranged between 1 and 13 species (Figure 5). *P. austroarabica* and *P. curviflorus* were found parasitizing on 13 and 5 of the host plants, respectively. *Vachellias* was the favored host for *P. austroarabica* and *P. curviflorus* with 83.4% and 73.5% infestations, respectively. *C. campestris*

was found on *Nicotiana glauca* (2%), *Pluchea dioscoridis* (28%), and *Pulicaria undulata* (70%), while *L. deflersii* was reported on *Vachellia tortilis* subsp. *tortilis* (69.6%), *Tamarix senegalensis* (21.7%), and *Searsia retinorrhoea* (8.7%). The genus *Orobanche* was represented by two species, including *O. cernua* on only one host species (*Rumex nervosus*) and *O. mutelii* on *Bidens biternata* (22.1%) and *R. nervosus* (77.9%). *P. acaciae* was found parasitizing *Senegalia asak* (58.1%), *Vachellia tortilis* subsp *raddiana* (32.3), and *Tamarix aphylla* (9.7%). *V schimperi* was found attacking two different host species, *Ziziphus spina-christi* (92.9%) and *T. aphylla* (7.1%).

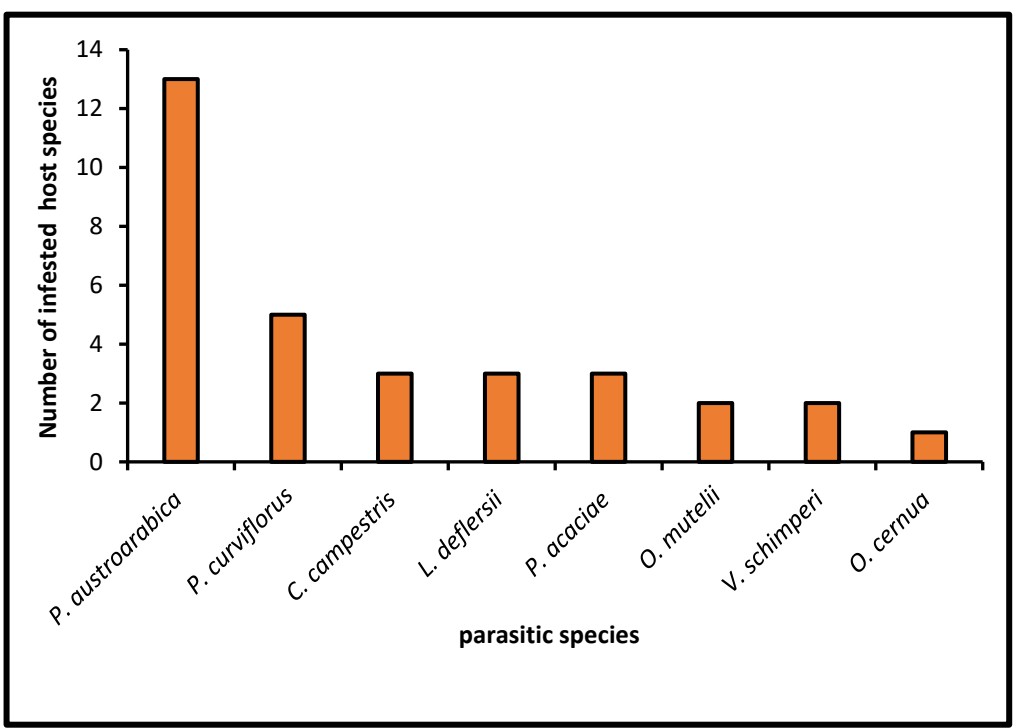

**Figure 5.** The number of host species infected by parasitic species in the study region.

### 3.2. Chorological Analysis of Parasitic Plants

The chorological analysis revealed that the identified parasitic species belonged to two major groups: biregional and monoregional (Table 1). Five species, accounting for 62.5% of the total number of species reported in the area, belonged to biregional taxa. Saharo-Arabian shared with Sudano-Zambezian had the highest share of species with three species (*L. deflersii*, *V. schimperi*, and *P. curviflorus*), followed by Saharo-Arabian /Irano-Turanian with two species. On the other hand, three species representing 37.5% of the parasitic species were monoregional. The three plants were native to the American (*C. campestris*), Saharo-Arabian (*P. austroarabica*), and Sudano-Zambezian (*P. acaciae*) regions. The mechanisms of seed dispersal for the eight parasitic plants are represented in Figure 6. Zoochory was the most prevalent seed dispersion method (six species), followed by anempchory (two species).

**Table 1.** The number of parasitic and host species and their relevant percentages (%) classified into different regional chorotypes in the study region.

| Chorotype | Parasitic Species | | Host Species | |
|---|---|---|---|---|
| | Number of Species | Percentage (%) | Number of Species | Percentage (%) |
| Monoregional | | | | |
| AM | 1 | 12.5 | - | - |
| SA-AR | 1 | 12.5 | - | - |
| SU-ZA | 1 | 12.5 | - | - |
| NEO | - | - | 1 | 4.35 |
| PAN | - | - | 1 | 4.35 |
| Biregional | | | | |
| SA-AR + SU-ZA | 3 | 37.5 | 18 | 78.3 |
| SA-AR + IR-TR | 2 | 12.5 | - | - |
| SU-ZA + IR-TR | | | 1 | 4.35 |
| Pluriregional | | | | |
| SA-AR + SU-ZA + ME | - | - | 1 | 4.35 |
| SA-AR + ME + TRO | - | - | 1 | 4.35 |
| Total | 8 | 100 | 23 | 100 |

ME: Mediterranean; IR-TR: Irano-Turanian; SA-AR: Saharo-Arabian; SU-ZA: Sudano-Zambezian; AM: American; TRO: Tropical; NEO: Neotropical; PAN: Pantropical.

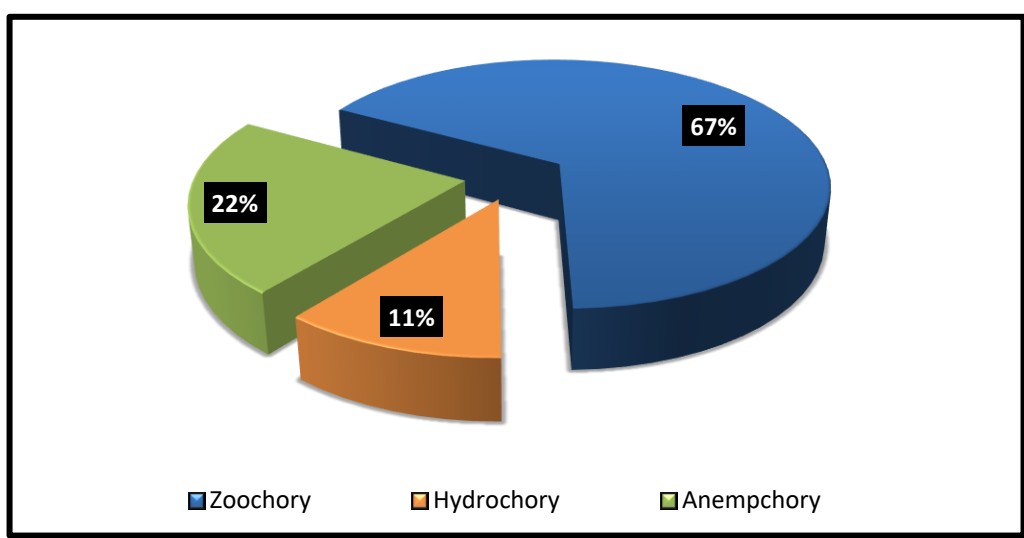

**Figure 6.** Dispersal mechanisms for parasitic species in the study region.

*3.3. Diversity of Parasitic Species*

The IVI values of the parasitic species were in the range of 107.28 to 10.21 (Table 2). The most important parasitic plants were *P. austroarabica*, *P. curviflorus*, and *V. schimperi*, with importance values of 107.28, 51.01 and 50.98, respectively, whereas *O. mutelii* was the least important species in the study region with an importance value of 10.21.

**Table 2.** Density (D), relative density (RD), frequency (F), relative frequency (RF), abundance (A), relative abundance (RA), and importance value index (IVI) for the parasitic species in the study region.

| Parasite Species | D | RD | F | RF | A | RA | IVI |
|---|---|---|---|---|---|---|---|
| *Cuscuta campestris* Yunck. | 0.156 | 4.28 | 10.94 | 8.38 | 1.43 | 8.36 | 21.02 |
| *Loranthella deflersii* (Tiegh.) S.Blanco & C.E.Wetzel | 0.072 | 1.98 | 6.56 | 5.02 | 1.1 | 6.43 | 13.43 |
| *Orobanche cernua* Loefl. | 0.034 | 0.93 | 2.81 | 2.15 | 1.22 | 7.13 | 10.21 |
| *Orobanche mutelii* F.W.Schultz | 0.269 | 7.38 | 19.06 | 14.59 | 1.41 | 8.25 | 30.22 |
| *Phragmanthera austroarabica* A.G.Mill. & J.A.Nyberg | 1.825 | 50.08 | 37.81 | 28.95 | 4.83 | 28.25 | 107.28 |
| *Plicosepalus acaciae* (Zucc.) Wiens & Polhill | 0.097 | 2.66 | 9.06 | 6.94 | 1.07 | 6.26 | 15.86 |
| *Plicosepalus curviflorus* Tiegh. | 0.6 | 16.47 | 29.69 | 22.73 | 2.02 | 11.81 | 51.01 |
| *Viscum schimperi* Engl. | 0.591 | 16.22 | 14.69 | 11.25 | 4.02 | 23.51 | 50.98 |
| Total | 3.644 | 100 | 130.62 | 100 | 17.1 | 100 | 300.01 |

*3.4. Floristic Composition of Host Species*

In the dry zone, 104 individuals representing 13 species and 5 families were found in site 1, whereas 102 individuals representing 13 species and 6 families were recorded in site 2 (Figures 7 and 8). In the wet zone, 148 individuals were recorded in site 3 representing 19 species and 10 families, whereas 164 individuals, 20 species, and 9 families were documented in site 4. The most represented host plant families in all the sites were Fabaceae, with 6, 6, 7, and 8 species in sites 1, 2, 3, and 4, respectively. Asteraceae was the second highest number with three species in each site. In each of the following families, only one species was identified: Apocynaceae, Barbeyaceae, Oleaceae, Polygonaceae, Rhamnaceae, Scrophulariaceae, and Solanaceae. In the surveyed sites, the perennial parasitic plants (91.3%) were considered the predominant type over the annual (8.7%) species (Table A2). Among the host species, 20 species were indigenous (87%) and three species were exotic (13%).

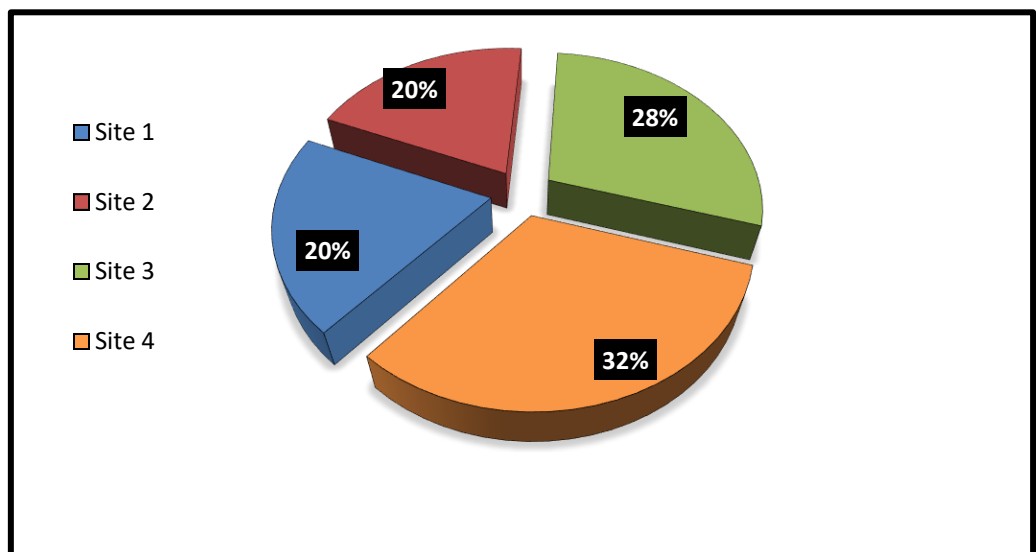

**Figure 7.** Percentage of the total number of individual host species at each site. Dry zone (site 1: 1300–1500, site 2: 1501–1700 m.a.s.l.); wet zone (site 3: 1800–2100, site 4: 2101–2400 m.a.s.l.).

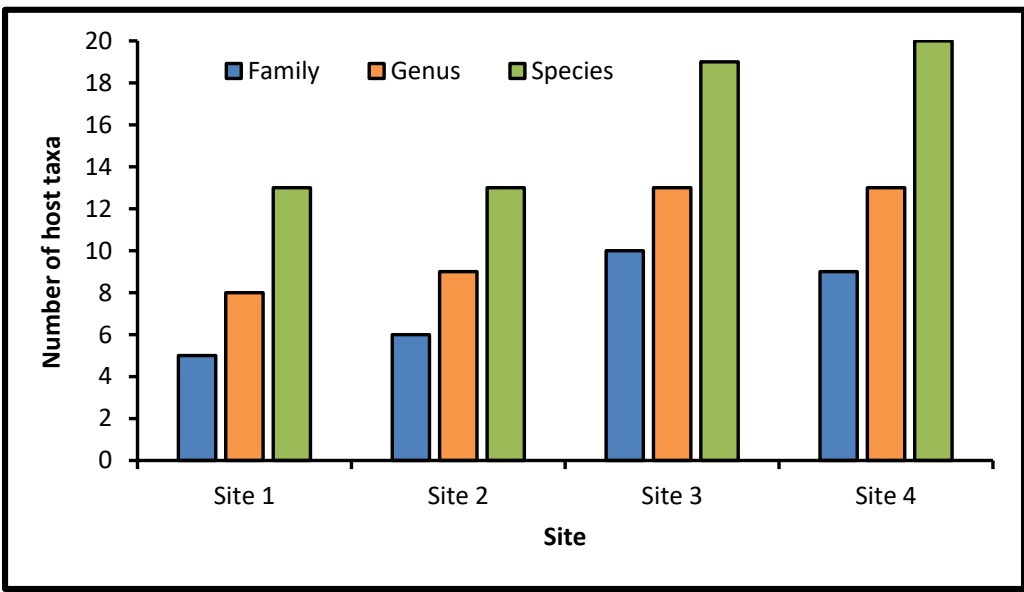

**Figure 8.** The number of families, genera, species, and total individuals of host plants in each site. Dry zone (site 1: 1300–1500, site 2: 1501–1700 m.a.s.l.); wet zone (site 3: 1800–2100, site 4: 2101–2400 m.a.s.l.).

Eighteen species were detected in each site of the study region (Table A2). *Vachellia origena*, *Barbeya oleoides*, *Buddleja polystachya*, and *Ficus carica* were found only in the wet zone, while *Olea europaea* subsp. *cuspidata* was absent in site 1. It also indicated that six host species were more affected by parasitic species (≥20%) and 17 species were less affected (<20%). The percentage of infested trees by parasitic species showed that *Vachellia flava* was the highest-infested tree (*P. austroarabica* = 47.5%), followed by *Z. spina-christi* (*V. schimperi* = 30.1%, *P. austroarabica* = 7.4%), *V. tortilis* subsp. *tortilis* (*L. deflersii* = 26.7%), *Vachellia gerrardii* (*P. austroarabica* = 19.8%, *P. curviflorus* = 5.8%), and *V. tortilis* subsp. *raddiana* (*P. austroarabica* = 13.8%, *P. acacia* = 5.3%, *P. curviflorus* = 5.3%).

The plant life form analysis demonstrated that three life forms of host species were reported in the four sites (Figure 9). Phanerophytes were the most common life form (19 species = 83%), followed by Therophytes (3 species = 13%), while Hemicryptophytes were represented by one species. The overall diversity of native trees was higher when compared with that of non-native tree species. Additionally, the number of perennial host plants in all sites were 21 species, whereas the annual plants were 2 species.

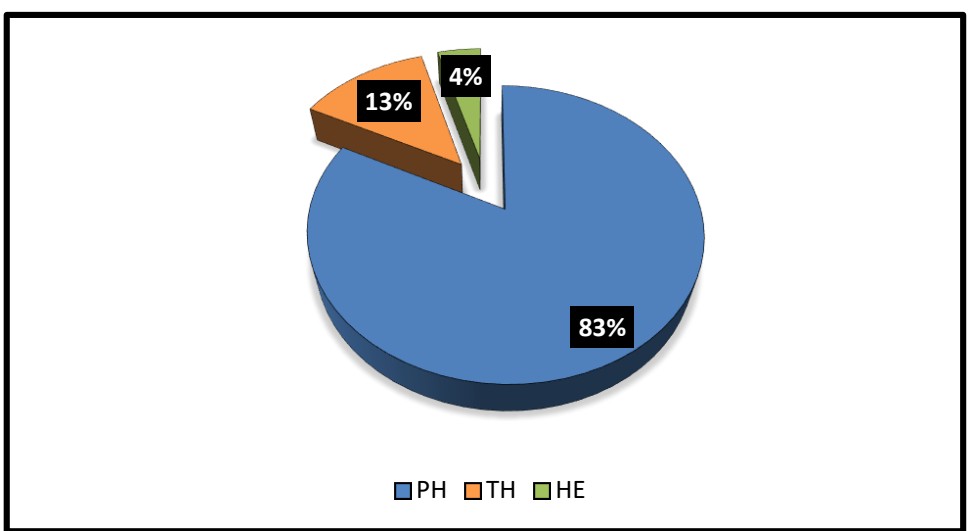

**Figure 9.** Life form categories of the host species in the study region. TH: Therophytes; PH: Phanerophytes; HE: Hemicryptophytes.

### 3.5. Chorological Analysis of Host Plants

The chorological analysis categorized the host species into three main categories, pluriregional, biregional, and monoregional. Most of the reported host plants (19 species = 83%) in the region belonged to the biregional group (Table 1). The biregional group had two chorotypes, with the Saharo-Arabian/Sudano-Zambezian chorotypes having the greatest number of species (18 species), while the Sudano-Zambezian/ Irano-Turanian included only one species, *T. aphylla*.

Only two pluriregional species were reported in the studied sites. Saharo-Arabian/ Sudano-Zambezian/Mediterranean was represented by one species (*T. senegalensis*), and Saharo-Arabian/Mediterranean/Tropical also was represented by one species (*Ficus palmata*). The monoregional elements were represented by two species. The detected monoregional species fall under two chorotypes: Neotropical (represented by one species, *B. biternata*) and Pantropical (represented by one species, *N. glauca*).

### 3.6. Diversity of Host Species

The calculated importance value indices (IVI) for the host species ranged between 32.44 and 2.31%. As shown in Table 3, seven plant species had the highest levels of IVI across the studied region. The dominant species were *N. glauca* (32.44%), *P. undulata* (28.24%), *S. asak* (27.14%), *Vachellia etbaica* (19.66%), *V. gerrardii* (19.09%), *O. europaea* subsp. *cuspidata* (18.59%), and *B. biternata* (18.21%). On the other hand, the least important species found in the study region were *V. tortilis* subsp. *tortilis* (4.68%), *Pistacia falcata* (4.36%), *T. senegalensis* (4.25%), *B. polystachya* (3.04%), and *F. carica* (2.31%). The importance value indices (IVI) for these species were 4.68%, 4.36%, 4.25%, 3.04%, and 2.31%, in order.

**Table 3.** Density (D), relative density (RD), frequency (F), relative frequency (RF), abundance (A), relative abundance (RA), and importance value index (IVI) for the host plant species in the study region.

| Host Species | D | RD | F | RF | A | RA | IVI |
|---|---|---|---|---|---|---|---|
| *Barbeya oleoides* Schweinf. | 0.28 | 1.61 | 13.13 | 3.2 | 2.14 | 2.22 | 7.03 |
| *Bidens biternata* (Lour.) Merr. & Sherff | 1.11 | 6.38 | 13.44 | 3.27 | 8.26 | 8.56 | 18.21 |
| *Buddleja polystachya* Fresen. | 0.07 | 0.4 | 4.06 | 0.99 | 1.62 | 1.68 | 3.07 |
| *Calotropis procera* (Aiton) W.T.Aiton | 0.49 | 2.82 | 19.06 | 4.64 | 2.59 | 2.68 | 10.14 |
| *Ficus carica* L. | 0.16 | 0.92 | 1.56 | 0.38 | 1 | 1.04 | 2.34 |
| *Ficus Palmata* Forssk. | 0.61 | 3.51 | 24.38 | 5.93 | 2.5 | 2.59 | 12.03 |
| *Nicotiana glauca* Graham | 2.17 | 12.47 | 13.44 | 3.27 | 16.12 | 16.71 | 32.45 |
| *Olea europaea* subsp. *cuspidata* (Wall. & G.Don) Cif. | 1.18 | 6.78 | 33.44 | 8.14 | 3.53 | 3.66 | 18.58 |
| *Pistacia falcata* Becc. ex Martelli. | 0.13 | 0.75 | 8.44 | 2.05 | 1.52 | 1.58 | 4.38 |
| *Pluchea dioscoridis* (L.) DC. | 0.36 | 2.07 | 8.44 | 2.05 | 4.25 | 4.4 | 8.52 |
| *Pulicaria undulata* (L.) C.A.Mey. | 1.79 | 10.29 | 12.5 | 3.04 | 14.35 | 14.87 | 28.2 |
| *Rumex nervosus* Vahl | 0.68 | 3.91 | 10 | 2.43 | 6.75 | 7 | 13.34 |
| *Senegalia asak* (Forssk.) Kyal. & Boatwr. | 1.76 | 10.11 | 56.56 | 13.76 | 3.12 | 3.23 | 27.1 |
| *Searsia retinorrhoea* (Steud. ex Oliv.) Moffett | 0.94 | 5.4 | 5.94 | 1.45 | 1.579 | 1.64 | 8.49 |
| *Tamarix aphylla* (L.) H.Karst. | 0.73 | 4.2 | 17.81 | 4.33 | 4.11 | 4.26 | 12.79 |
| *Tamarix senegalensis* DC. | 0.1 | 0.57 | 3.75 | 0.91 | 2.67 | 2.77 | 4.25 |
| *Vachellia etbaica* (Schweinf.) Kyal. & Boatwr. | 1.3 | 7.47 | 33.75 | 8.21 | 3.84 | 3.98 | 19.66 |
| *Vachellia flava* (Forssk.) Kyal. & Boatwr. | 0.25 | 1.44 | 11.56 | 2.81 | 2.16 | 2.24 | 6.49 |
| *Vachellia gerrardii* (Benth.) P.J.H.Hurter | 1.24 | 7.13 | 33.13 | 8.1 | 3.76 | 3.9 | 19.13 |
| *Vachellia origena* (Hunde) Kyal. & Boatwr. | 0.8 | 4.6 | 23.75 | 5.78 | 3.38 | 3.5 | 13.88 |
| *Vachellia tortilis* subsp. *raddiana* (Savi) Kyal. & Boatwr. | 0.48 | 2.76 | 22.5 | 5.48 | 2.11 | 2.19 | 10.43 |
| *Vachellia tortilis* subsp. *tortilis* | 0.09 | 0.52 | 2.81 | 0.68 | 3.33 | 3.45 | 4.65 |
| *Ziziphus spina-christi* (L.) Willd. | 0.68 | 3.91 | 37.5 | 9.13 | 1.8 | 1.87 | 14.91 |
| Total | 17.4 | 100.02 | 410.95 | 100 | 96.489 | 100 | 300 |

## 4. Discussion

Studies on the distribution and diversity of parasitic plants in natural plant communities are lacking for many regions of Saudi Arabia. To my knowledge, this is the first attempt to study the ecology and diversity patterns of these highly specialized plants in the Al-Baha region, in the southwest part of Saudi Arabia. Previous studies focused only on the general floristic compositions and the structure of plant communities in specific habitats. The current study recorded eight species of the parasitic plants from four families. There were 34 parasitic species belonging to eight families recorded in the flora of Saudi Arabia [39]. However, the number of parasitic plants in Saudi Arabia is much lower than that of Nepal (151) [40], Turkey (146), and China (678 species) [41]. This variation could be attributed to host availability and abiotic stress, which are the key factors affecting the distribution and abundance of parasitic plants or other ecological factors. Furthermore, the success of parasitic plants under adverse conditions is highly dependent on host selection [11].

In the southern Andes at high altitudes, Amico et al. [42] demonstrated that the Andean-Patagonian Forest is rich in parasitic mistletoes. This finding agreed with our result which showed that the high-altitude sites had high numbers of parasites as compared with the low-altitude sites. On the other hand, the abundance of parasitic plants at high-altitude sites could be related to the richness of host plants in these sites. However, Hechinger and Lafferty [43] revealed that high host diversity can assist the diversity of parasitic plants.

In the current study, the most dominant family of parasitic species was Loranthaceae with four species, followed by Orobanchaceae with two species. Stem hemiparasites were the dominant parasitic group with five species, while only two species belonging to root holoparasitic plants were detected in the studied sites. This result was consistent with the fact that the majority of parasitic species are hemiparasites [44]. Moreover, hemiparasitic plants had morphological characteristics with a wide range of host interaction [44], often parasitizing multiple plant species [45]. The diversity and distribution of both hemiparasitic and holoparasitic plants across the study region could be attributed to the effects of

topography and climate factors, especially on the growth of seedlings and germination of parasitic species.

Drought stress is another factor that affects the growth and distribution of hemiparasitic and holoparasitic plants [11]. The deficiency of water availability negatively affects the development and the growth rate of seeds of the root parasites *Orobanche crenata* [46], *Striga hermonthica*, and *Alectra vogelii* [47].

Our results revealed that *P. austroarabica* and *P. curviflorus* parasitized 13 and 5 different host plants, respectively. *Vachellias* was the most favored host for *P. austroarabica* and *P. curviflorus* with 83.4% and 73.5% infestation, respectively. This might be due to the dominance of *Vachellias* trees, which were the highest-recorded host species in the study area. This finding was consistent with the results of Migahid [48], who stated that *P. austroarabica* and *P. curviflorus* are common parasites of *Vachellias* in Saudi Arabia. On the other hand, Amico et al. [42] reported 12 mistletoes that parasitize 43 species (from 23 families) out of 185 woody species in the Andean-Patagonian Forest. Moreover, eight of these mistletoes are specialists with restricted host range and the remaining are generalists. In the wet zone, *C. campestris* parasitized *N. glauca*, which is one of the most invasive species in Saudi Arabia. Previous studies in Chian indicated that the genus *Cuscuta* suppress the invasive plants *Ipomoea cairica*, *Mikania micrantha*, *Wedelia trilobata*, *Solidago canadensis*, *Bidens pilosa*, and *Humulus scandens* [49,50]. As a result, it is expected that the reported *C. campestris* in the wet sites will suppress the widespread *N. glauca*. Těšitel et al. [10] demonstrated that some native parasitic plant species could be used to repress plant invasions and help restore biodiversity. Thus, it is possible that the indigenous parasites identified in the study region could benefit this habitat by controlling invading plants.

The chorological analysis indicated that parasitic species from the Saharo-Arabian and Sudano-Zambezian regions dominated the region. According to Zohary [34], the vegetation of Saudi Arabia belongs to that of the Saharo-Arabian region. Abdel Khalik et al. [51] showed that the Saharo-Arabian and Sudano-Zambezian species were the best biomarkers of arid climate. Seed dispersal of parasitic species revealed that zoochory was the main dispersal mechanism for six species, whereas anemochory was the dispersion mode of both *Orobanche* species.

After feeding on mistletoe fruits, generalist birds regurgitate or defecate the sticky seeds which paste onto woody branches [52]. Birds dispersing mistletoe seeds demonstrate the great degree of coevolution between them [53], which also has an important function in pollination [52]. Other studies have reported that birds are the primary seed dispersers, with some seeds being dispersed by wind or hydraulic explosives [54,55].

As observed from the IVI analysis, *P. austroarabica* had the highest importance value (107.28) as compared with other parasitic species. Magray et al. [56] reported that the variation in the IVI of species might be caused by predominant environmental factors. Moreover, the difference in IVI among the sites may be due to the composition of plant species, human activities, and local ecological factors [57].

The number of host species, genera, and families varied across the surveyed sites. In the dry zone, 206 individuals from 14 species and 6 families were recorded, whereas 312 individuals from 22 species and 12 families were recorded in the wet zone. Vegetation analysis of the host species demonstrated that Fabaceae (seven species) and Asteraceae (three species) were the two top host species-rich families. These results were consonant with previous studies on the Saudi Arabian flora [58,59]. However, Fabaceae and Asteraceae were notified in the flora of the Mediterranean, North Africa, eastern Ethiopia, and northern Zambia [59]. Anacardiaceae, Moraceae, and Tamaricaceae each had only two species (8.7%), whereas seven families (30.43%) were represented by just one species. As compared with desert vegetation, the majority of plant species in Saudi Arabia are members of a few families and about 58% of the families were represented by a single species. Al Nafie [58] recorded that 24.2% of the families in Saudi Arabia's flora are represented by one species per family. In the surveyed region, the perennial types (91.3%) prevailed over the annuals (8.7%). The host plants included 20 indigenous (87%) and 3 (13%) exotic species.

In southwest China, Luo et al. [60] found only three hosts out of 88 species in the tropical forest that were parasitized by *Dendrophthoe pentandra*, which indicates that the abundance and host richness of species did not explain the frequency of infections at the sampling unit. Zhang et al. [41] reported that the abundance of parasitic plants in a particular site is usually determined by environmental (altitude, area, longitude, and latitude) and biological (dispersal vector and host availability) factors. Host plants' diversity and different kinds of habitats can also influence the spreading and density of parasitic species [61]. Further, the development and growth of parasitic species were closely correlated to the features of their hosts [62,63]. Roxburgh and Nicolson [64] demonstrated that the age of the host species was correlated with the number of mistletoes, so a high number of mistletoe clusters enhanced the probability that additional mistletoe seeds could germinate on that host tree, resulting in the growth of more mistletoe clusters.

The obtained results demonstrated that parasitic species infection varied among the tree species, with *V. flava* being the highest infested tree by *P. austroarabica*, followed by *Z. spina-christi* and *V. tortilis* subsp. *tortilis*. The woody parasites *P. acaciae* and *P. curviflorus* are widespread parasites of *Vachellia* [48]. Al-Rowaily et al. [24] reported a high infection incidence of *P. curviflorus* in different species of *Vachellia* genus in Saudi Arabia's southern and western areas, resulting in ecosystem degradation and loss of diversity and soil nutrients. Other studies demonstrated that mistletoe preferentially infects massive trees over small ones [64,65]. Therefore, the difference in infection rate could possibly be correlated to host size [66]. This can be linked to the fact that dispersal birds prefer larger trees for resting and feeding [67]. Further, some parasitic species preferred woody hosts, which may be consistent with the perennial life form and hemiparasitic nature [68,69]. On the other hand, some herbal species may work as bridging hosts to enable seedlings to live long enough to grow onto nearby shrubs or trees [70].

In the present investigation, three life forms were noticed, with Phanerophytes being the most prominent (19 species = 83%), followed by Therophytes (three species = 13%). This result agreed with the findings that were reported by Abbas et al. [59] and Elkordy et al. [71] in different regions of the Kingdom. The dominance of perennial species reflects the characterization of the vegetation in the studied region. This may be caused by low precipitation and a long dry season, which is not sufficient for the growth of annual species [72,73]. Further, perennial plants can acclimate to the extreme ecological conditions of the area.

According to the chorological analysis results, the biregional elements of the Saharo-Arabian/Sudano-Zambezian chorotypes had the most dominant share of host species by 18 species. Comparable findings were found in several investigations in Saudi Arabian flora [59,71,73].

Native plants showed higher levels of species richness and diversity when compared to non-native plants. The abiotic variables such as the amount of rainfall and the aridity of the area are the most affected factors on the dominance of perennial species [51,73,74]. This is a prominent characteristic of the vegetation of the Al-Baha region since perennial species might be more resistant to climatological change than annual species.

Based on the IVI analysis, the most important host species were *N. glauca*, *P. undulata*, *S. asak*, *V. etbaica*, *V. gerrardii*, *O. europaea* subsp. *cuspidata*, and *B. biternata*. Importance value indices can be applied in vegetation analysis to determine the order of species conservation. Species that have low importance values must be afforded higher conservation priority than species with high importance values [75].

The variations in the obtained results among the studied sites could be mainly due to the elevation gradient. However, Al-Robai et al. [21] revealed the effects of elevation gradient on structure and diversity of plant communities along the Alabna escarpment in the Al-Baha area. Further, Al-Namazi et al. [22] demonstrated that the high-attitude area of the Al-Baha region has a wide range of plant diversity.

### 5. Study Limitations

An important study limitation to consider is the inability to compare, monitor, and assess the distribution and diversity of parasitic plants in the surveyed region due to the lack of previous research on these species. The study focuses on the elevation gradient as the major element that may influence the distribution and diversity of parasite and host plants, whereas other environmental variables may have an impact on the structure and dynamics of these plants. It should be kept in mind that this study was conducted in a rather small and restricted location. Future research will expand on other habitats to cover more geographical regions across the country to fully examine the main factors that affect the diversity of parasitic plants and their hosts in the selected studied regions. This will be performed by employing more environmental parameters.

### 6. Conclusions

Vegetation analysis of density, abundance, and frequency revealed a clear variation of both parasite and host plants along the elevation gradient. Elevation, unique topography, and climatic conditions of the study region could possibly be responsible for these variations as well as other environmental factors. Eight parasites and 23 host species were recorded across the surveyed four sites. *Phragmanthera austroarabica* was widely spread throughout the sites and was reported in most of the studied sampling units. It represented half of the total individuals in the region and had the highest infection rate, followed by *P. curviflorus*. Half of the parasitic species (50%) found in the region belonged to Loranthaceae family. The highest alpha diversity of the reported hosts (20) was detected in site 4, while the lowest (13) was reported in sites 1 and 2. Fabaceae was the most common host plant family, and seven families were represented by only one host species. The most infested tree was *Vachellia flava*, followed by *Z. spina-christi*, and *V. tortilis* subsp. *raddiana* was the least infected species. Perennial, therophyte, and bioregional elements were the dominant host species in both the dry and wet zones. It is suggested that *Cuscuta campestris* can be exploited as a biological control tool to reduce the spread of *N. glauca* in the Al-Baha region. The majority of the recorded parasite and host plants were indigenous to Saudi Arabia. Further work at the local or broad geographical regions is recommended all over the country to better understand and document the diversity, structure, and dynamics of parasitic plants.

**Funding:** This research received no external funding.

**Institutional Review Board Statement:** Not applicable.

**Data Availability Statement:** The data presented in this study are available in this article.

**Conflicts of Interest:** The author declares no conflict of interest.

## Appendix A

**Table A1.** List of parasitic species recorded in the studied region with their families, spatial distribution, kind of parasite, origin, habit, life span, chorotype, dispersal type, infested plant organ, hosts, and percentage of infestation.

| Family/Parasite | Spatial Distribution | Kind of Parasite | Origin | Habit | Life Span | Chorotype | Dispersal Type | Infested Plant Organ | Host | INFC (%) |
|---|---|---|---|---|---|---|---|---|---|---|
| **Convolvulaceae** | | | | | | | | | | |
| *Cuscuta campestris* Yunck. | 1.2.3.4 | Holop. | EXO | Herb | Ann. | AM | ZO + HY | Stem | *Nicotiana glauca* Graham | 2 |
| | | | | | | | | | *Pluchea dioscoridis* (L.) DC. | 28 |
| | | | | | | | | | *Pulicaria undulata* (L.) C.A.Mey. | 70 |
| **Loranthaceae** | | | | | | | | | | |
| *Loranthella deflersii* (Tiegh.) S.Blanco & C.E.Wetzel | 3.4 | Hmip. | IND | Shrub | Per. | SA-AR + SU-ZA | ZO | Stem | *Searsia retinorrhoea* (Steud. ex Oliv.) Moffett | 8.7 |
| | | | | | | | | | *Tamarix senegalensis* DC. | 21.7 |
| | | | | | | | | | *Vachellia tortilis* subsp. *tortilis* | 69.6 |
| *Phragmanthera austroarabica* A.G.Mill. & J.A.Nyberg | 1.2.3.4 | Hmip. | IND | Shrub | Per. | SA-AR | ZO | Stem | *Barbeya oleoides* Schweinf. | 2.6 |
| | | | | | | | | | *Buddleja polystachya* Fresen. | 0.17 |
| | | | | | | | | | *Calotropis procera* (Aiton) W.T.Aiton | 0.17 |
| | | | | | | | | | *Ficus carica* L. | 3.8 |
| | | | | | | | | | *Ficus Palmata* Forssk. | 0.17 |
| | | | | | | | | | *Olea europaea* subsp. *cuspidata* (Wall. & G.Don) Cif. | 3.1 |
| | | | | | | | | | *Pistacia falcata* Becc. ex Martelli. | 0.51 |
| | | | | | | | | | *Tamarix aphylla* (L.) H.Karst. | 2.6 |
| | | | | | | | | | *Vachellia flava* (Forssk.) Kyal. & Boatwr. | 11.5 |
| | | | | | | | | | *Vachellia gerrardii* (Benth.) P.J.H.Hurter | 33.7 |
| | | | | | | | | | *Vachellia origena* (Hunde) Kyal. & Boatwr. | 27.6 |
| | | | | | | | | | *Vachellia tortilis* subsp. *raddiana* (Savi) Kyal. & Boatwr. | 10.6 |
| | | | | | | | | | *Ziziphus spina-christi* (L.) Willd. | 3.6 |

**Table A1.** *Cont.*

| Family/Parasite | Spatial Distribution | Kind of Parasite | Origin | Habit | Life Span | Chorotype | Dispersal Type | Infested Plant Organ | Host | INFC (%) |
|---|---|---|---|---|---|---|---|---|---|---|
| *Plicosepalus acaciae* (Zucc.) Wiens & Polhill | 1.2.3.4 | Hmip. | IND | Shrub | Per. | SU-ZA | ZO | Stem | *Senegalia asak* (Forssk.) Kyal. & Boatwr. | 58.1 |
| | | | | | | | | | *Tamarix aphylla* (L.) H.Karst. | 9.7 |
| | | | | | | | | | *Vachellia tortilis* subsp. *raddiana* (Savi) Kyal. & Boatwr. | 32.3 |
| *Plicosepalus curviflorus* Tiegh. | 1.2.3.4 | Hmip. | IND | Shrub | Per. | SA-AR + SU-ZA | ZO | Stem | *Senegalia asak* (Forssk.) Kyal. & Boatwr. | 21.4 |
| | | | | | | | | | *Tamarix aphylla* (L.) H.Karst. | 5.2 |
| | | | | | | | | | *Vachellia etbaica* (Schweinf.) Kyal. & Boatwr. | 31.8 |
| | | | | | | | | | *Vachellia gerrardii* (Benth.) P.J.H.Hurter | 26.6 |
| | | | | | | | | | *Vachellia tortilis* subsp. *raddiana* (Savi) Kyal. & Boatwr. | 15.1 |
| **Orobanchaceae** | | | | | | | | | | |
| *Orobanche mutelii* F.W.Schultz | 1.2.3.4 | Holop. | IND EXO | Herb Herb | Ann. Ann. | SA-AR + IR-TR Med + IR-TR | AN AN | Root | *Bidens biternata* (Lour.) Merr. & Sherff | 22.1 |
| | | | | | | | | | *Rumex nervosus* Vahl | 77.9 |
| *Orobanche cernua* Loefl. | 3.4 | Holop. | IND | Herb | Ann. | SA-AR + IR-TR | AN | Root | *Rumex nervosus* Vahl | 100 |
| **Santalaceae** | | | | | | | | | | |
| *Viscum schimperi* Engl. | 1.2.3.4 | Hmip. | IND | Shrub | Per. | SA-AR + SU-ZA | ZO | Stem | *Tamarix aphylla* (L.) H.Karst. | 7.1 |
| | | | | | | | | | *Ziziphus spina-christi* (L.) Willd. | 92.9 |

Kinds of parasite abbreviations: Holop.: holoparasite; Hmip.: Hemiparasitic; INFC: infestation. Life span abbreviations: Per.: perennial; Ann.: annual. Origin abbreviations: IND: indigenous; EXO: exotic. Chorotype abbreviations: ME: Mediterranean; IR-TR: Irano-Turanian; SA-AR: Saharo-Arabian; SU-ZA: Sudano-Zambezian; AM: American. Dispersal type abbreviations: ZO: zoochory; HY: hydrochory; AN: anempchory.

**Table A2.** List of host species with their families, spatial distribution, life form, life span, origin, chorotypes, parasitic species, and the percentage of the infestation.

| Family/Host | Spatial Distribution | Life Form | Life Span | Origin | Chorotype | Parasitic Species | Infestation (%) |
|---|---|---|---|---|---|---|---|
| **Asteraceae** | | | | | | | |
| *Bidens biternata* (Lour.) Merr. & Sherff | 1.2.3.4 | TH | Ann. | IND | NEO | *Orobanche mutelii* F.W.Schultz | 2 |
| *Pluchea dioscoridis* (L.) DC. | 1.2.3.4 | TH | Per. | IND | SA-AR + SU-ZA | *Cuscuta campestris* Yunck | 11.3 |
| *Pulicaria undulata* (L.) C.A.Mey. | 1.2.3.4 | HE | Per. | IND | SA-AR + SU-ZA | *Cuscuta campestris* Yunck | 6.3 |
| **Anacardiaceae** | | | | | | | |
| *Pistacia falcata* Becc. ex Martelli. | 1.2.3.4 | PH | Per. | IND | SA-AR + SU-ZA | *Phragmanthera austroarabica* A.G.Mill. & J.A.Nyberg | 2.4 |
| *Searsia retinorrhoea* (Steud. ex Oliv.) Moffett | 1.2.3.4 | PH | Per. | IND | SA-AR + SU-ZA | *Loranthella deflersii* (Tiegh.) S.Blanco & C.E.Wetzel | 3.3 |
| **Apocynaceae** | | | | | | | |
| *Calotropis procera* (Aiton) W.T.Aiton | 1.2.3.4 | PH | Per. | IND | SA-AR + SU-ZA | *Phragmanthera austroarabica* A.G.Mill. & J.A.Nyberg | 0.6 |
| **Barbeyaceae** | | | | | | | |
| *Barbeya oleoides* Schweinf. | 3,4 | PH | Per. | IND | SA-AR + SU-ZA | *Phragmanthera austroarabica* A.G.Mill. & J.A.Nyberg | 2.2 |
| **Fabaceae** | | | | | | | |
| *Senegalia asak* (Forssk.) Kyal. & Boatwr. | 1.2.3.4 | PH | Per. | IND | SA-AR + SU-ZA | *Plicosepalus acaciae* (Zucc.) Wiens & Polhill<br>*Plicosepalus curviflorus* Tiegh. | 2<br>3.4 |
| *Vachellia etbaica* (Schweinf.) Kyal. & Boatwr. | 1.2.3.4 | PH | Per. | IND | SA-AR + SU-ZA | *Plicosepalus curviflorus* Tiegh. | 8.2 |
| *Vachellia flava* (Forssk.) Kyal. & Boatwr. | 1.2.3.4 | PH | Per. | IND | SA-AR + SU-ZA | *Phragmanthera austroarabica* A.G.Mill. & J.A.Nyberg | 47.5 |
| *Vachellia gerrardii* (Benth.) P.J.H.Hurter | 1.2.3.4 | PH | Per. | IND | SA-AR + SU-ZA | *Phragmanthera austroarabica* A.G.Mill. & J.A.Nyberg<br>*Plicosepalus curviflorus* Tiegh. | 19.8<br>5.8 |
| *Vachellia origena* (Hunde) Kyal. & Boatwr. | 3,4 | PH | Per. | EXO | SA-AR + SU-ZA | *Phragmanthera austroarabica* A.G.Mill. & J.A.Nyberg | 16 |
| *Vachellia tortilis* subsp. *raddiana* (Savi) Kyal. & Boatwr. | 1.2.3.4 | PH | Per. | IND | SA-AR + SU-ZA | *Phragmanthera austroarabica* A.G.Mill. & J.A.Nyberg<br>*Plicosepalus acaciae* (Zucc.) Wiens & Polhill<br>*Plicosepalus curviflorus* Tiegh. | 13.8<br>5.3<br>5.3 |
| *Vachellia tortilis* subsp. *tortilis* | 1.2.3.4 | PH | Per. | IND | SA-AR + SU-ZA | *Loranthella deflersii* (Tiegh.) S.Blanco & C.E.Wetzel | 26.7 |

**Table A2.** *Cont.*

| Family/Host | Spatial Distribution | Life Form | Life Span | Origin | Chorotype | Parasitic Species | Infestation (%) |
|---|---|---|---|---|---|---|---|
| **Moraceae** | | | | | | | |
| *Ficus carica* L. | 3,4 | PH | Per. | EXO | SA-AR + SU-ZA | *Phragmanthera austroarabica* A.G.Mill. & J.A.Nyberg | 20 |
| *Ficus Palmata* Forssk. | 1.2.3.4 | PH | Per. | IND | SA-AR+ ME+TRO | *Phragmanthera austroarabica* A.G.Mill. & J.A.Nyberg | 1.5 |
| **Oleaceae** | | | | | | | |
| *Olea europaea* subsp. *cuspidata* (Wall. & G.Don) Cif. | 2.3.4 | PH | Per. | IND | SA-AR + SU-ZA | *Phragmanthera austroarabica* A.G.Mill. & J.A.Nyberg | 2.9 |
| **Polygonaceae** | | | | | | | |
| *Rumex nervosus* Vahl | 1.2.3.4 | TH | Ann. | IND | SA-AR + SU-ZA | *Orobanche mutelii* F.W.Schultz *Orobanche cernua* Loefl. | 9.7 4.2 |
| **Rhamnaceae** | | | | | | | |
| *Ziziphus spina-christi* (L.) Willd. | 1.2.3.4 | PH | Per. | IND | SA-AR + SU-ZA | *Phragmanthera austroarabica* A.G.Mill. & J.A.Nyberg *Viscum schimperi* Engl. | 7.4 30.1 |
| **Scrophulariaceae** | | | | | | | |
| *Buddleja polystachya* Fresen. | 3,4 | PH | Per. | IND | SA-AR + SU-ZA | *Phragmanthera austroarabica* A.G.Mill. & J.A.Nyberg | 4.8 |
| **Solanaceae** | | | | | | | |
| *Nicotiana glauca* Graham | 1.2.3.4 | PH | Per. | EXO | PAN | *Cuscuta campestris* Yunck | 0.1 |
| **Tamaricaceae** | | | | | | | |
| *Tamarix aphylla* (L.) H.Karst. | 1.2.3.4 | PH | Per. | IND | SU-ZA + IR-TR | *Phragmanthera austroarabica* A.G.Mill. & J.A.Nyberg *Viscum schimperi* Engl. *Plicosepalus acaciae* (Zucc.) Wiens & Polhill *Plicosepalus curviflorus* Tiegh. | 3.8 5.1 1.3 4.3 |
| *Tamarix senegalensis* DC. | 1.2.3.4 | PH | Per. | IND | SA-AR+SU-ZA+ME | *Loranthella deflersii* (Tiegh.) S.Blanco & C.E.Wetzel | 15.6 |

Life form abbreviations: PH: Phanerophytes; TH: Therophytes; HE: Hemicryptophytes. Life span abbreviations: Per.: perennial; Ann.: annual. Origin abbreviations: IND: indigenous; EXO: exotic. Chorotype abbreviations: ME: Mediterranean; NEO: Neotropical; PAN: Pantropical; IR-TR: Irano-Turanian; SA-AR: Saharo-Arabian; SU-ZA: Sudano-Zambezian; TRO: Tropical.

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
