# Peer review of "Ecology and Diversity of Angiosperm Parasites and Their Host Plants along Elevation Gradient in Al-Baha Region, Saudi Arabia"

_diversity, doi:10.3390/d15101065_

Round 1
Reviewer 1 Report
Review of “Ecology and Diversity of Angiosperm Parasites and Their Host 2 Plants along Elevation Gradients in Al-Baha Region, Saudi Arabia”.
This study reports on the distribution and diversity of angiosperm parasites along an elevation gradient. The author presents data on the diversity (richness and abundance) of plants at different sites. The results were overall well presented and discussed. However, the author does not report on how often sampling was done. Seasonal differences in both the parasite and host diversity would be expected, particularly regarding annual species. The author should clearly identify and state the assumptions and limitations of this study.
The author also speaks about different gradients (temperature and precipitation) but does not take the opportunity to explore this further by empirically testing the relationships between climate and species richness. Simply stating which species were found at the different sites is no different from what others have reported.
Below I provide some additional comments.
Abstract:
“Gradients” should be gradient because you mostly look at high and low elevations.
L13: Give the absolute number of species.
Remove “were”
L18: dry zone (low or higher elevation?)
L64: should be biological ecosystems.
L77: by the
L86: Were these once of sampling? Did you include seasonal sampling? This might have an impact on both your parasite and host species richness.
How did you distinguish between dry and wet zones?
Which criteria did you use to divide these regions?
Fig 1
Reduce the number of tick marks on the first map and the intervals of degrees.
L104: Explain What the original meaning of IVI is.
Fig 2
Reduce the size of the species names so that there is more space for the data.
Plot the species based on the number of individuals largest to smallest. so that P austroarabica is first.
Fig 3.
These pie charts do not need to be plotted and can be described in the results.
L143: What are you referring to here with 78 %?
Add percentage sign after 73.5%
L146: Genera are always in Italics.
L153-154: How did you do this analysis? It was not explained in the M&M. What is the meaning of these two regional terms?
Fig 4
This figure should be right after the text description. Follow the same suggestions as given in Fig. 3.
Table 2. pluriregional
Table 3. Authorities should be given when the species is first mentioned.
Fig 7
Change “Number of Taxa” to “Number of Host Taxa”.
L277: Ficus palmata.
L246: To my knowledge
L250: Where is this statement coming from? If it was not found in your work then you should cite someone.
L257: How high compared to your study?
L300: Do you have any information on which species are the dispersers?
L339-341: Remove text, as it is Results.
Minor editing of English language required
Author Response
Dear editor and reviewer,
I would like to thank the editor and reviewer for their thoughtful comments and helpful suggestions. I have addressed all issues indicated in the review report and I am hopeful that the revised version will meet the journal publication requirements. The following are detailed, point-by-point responses to the reviewer's comments with the associated amendments written in red text. The deleted words or sentences are crossed out with a strikethrough and highlighted in yellow throughout the revised manuscript.

Reviewer 2 Report
In general, the paper is well-written. It can be published after making the following corrections:
1- Write authority along with the scientific names of plants at their first citation in the text especially in the Abstract.
2- There are 27 references in Introduction. Reduce their number. Maximum 2 references are enough to support a statement. Also add one or 2 references of 2023.
3- Line 140-141: Rearrange the sentence to avoid repetition of the word "species".
4- Line 162: Write correctly as Fig. 5.
5- Fig. 4: On Y-axis, reduce the font size of "No. of infected host species" to fit it in the space.
6- Format Table 2 correctly. Also delete the "%" along with the digits. Writing once at the top is enough.
7- Tables and figures should be self explanatory. Write actual names of the sites either in the figure or as a foot note.
8- Line 199: Replace 18 with Eighteen. Cannot start a sentence with a digit.
9-Remove references from the Conclusion. It should not be like that of Discussion.

Author Response

(The authors gave the same response as above.)

Reviewer 3 Report
Short, well written paper. A few my suggestions and corrections:
In the introduction, I would propose to present what research on parasitic plants has already been carried out in Saudi Arabia. To show more of the novelty of the article.
I would suggest posting plate with photos of parasitic plants and their hosts.
The latin names of gerera should be written using italics.
Author Response

(The authors gave the same response as above.)

Round 2
Reviewer 2 Report
Revised version is very much improved. However, the following is not addressed.
1- Delete references from the conclusion. It should not be like that of Discussion. It should contain only a few concluding remarks based on the findings of the present study.
2- Number of references is too high. It is 85. I think around 50 or less number of references are enough for a research article.
Author Response

(The authors gave the same response as above.)
